# Effects of Chokeberries (*Aronia* spp.) on Cytoprotective and Cardiometabolic Markers and Semen Quality in 109 Mildly Hypercholesterolemic Danish Men: A Prospective, Double-Blinded, Randomized, Crossover Trial

**DOI:** 10.3390/jcm12010373

**Published:** 2023-01-03

**Authors:** Julie Sangild, Anne Faldborg, Cecilie Schousboe, Maja Døvling Kaspersen Fedder, Lars Porskjær Christensen, Astrid Komal Lausdahl, Eva Christensen Arnspang, Søren Gregersen, Henrik Byrial Jakobsen, Ulla Breth Knudsen, Jens Fedder

**Affiliations:** 1Department of Clinical Medicine, Aarhus University, DK-8200 Aarhus, Denmark; 2Laboratory of Reproductive Biology, Scientific Unit, Regional Hospital of Horsens, DK-8700 Horsens, Denmark; 3Department of Physics, Chemistry and Pharmacy, Faculty of Science, University of Southern Denmark, DK-5230 Odense, Denmark; 4Department of Green Technology, Faculty of Engineering, University of Southern Denmark, DK-5230 Odense, Denmark; 5Steno Diabetes Center Aarhus, DK-8200 Aarhus, Denmark; 6Byrial ApS, Blaesenborgvej 9, DK-4320 Lejre, Denmark; 7Department of Obstetrics and Gynecology, Fertility Clinic, Horsens Regional Hospital, DK-8700 Horsens, Denmark; 8Centre of Andrology, Fertility Clinic, Department D, Odense University Hospital, DK-5000 Odense, Denmark; 9Department of Clinical Medicine, University of Southern Denmark, DK-5000 Odense, Denmark

**Keywords:** Aronia supplementation, crossover clinical trial, oxidative stress, cytoprotective effects, glutathione, cholesterol

## Abstract

Background: Chokeberries (*Aronia* spp.) are known to exhibit both direct and indirect antioxidant properties and have been associated with beneficial effects on human health, including cardiovascular risk factors (inflammation, serum lipids, sugars, blood pressure), oxidative stress, and semen quality. This prospective, double-blinded, randomized, crossover clinical trial was conducted to elucidate the effects of Aronia supplementation on these health targets in mildly hypercholesterolemic men. Methods: The standardized Aronia supplementation comprised three wild *Aronia* spp. (*A. arbutifolia*, *A prunifolia* and *A. melanocarpa*) and the *Aronia* hybrid × *Sorbaronia mitschurinii* (standardized to 150 mg anthocyanins daily). Participants (*n* = 109) were healthy men with respect to all outcome targets except for the total cholesterol level (5.0–7.0 mM). Participants were randomized to supplementation with either Aronia or placebo for 90 days, followed by a wash-out period and lastly the complementary supplementation. Effects on the health parameters were compared among both the whole group of men and in subgroups according to age, body mass index (BMI), lifestyle, dietary habits, and serum glutathione levels at baseline. The study is registered in ClinicalTrials.gov.: NCT03405753. Results: Glutathione levels were significantly improved after 90 days intake of Aronia supplementation compared to placebo in the subgroup of men with a low level of glutathione at baseline (*p* = 0.038) and a high coffee intake (*p* = 0.045). A significant decrease in levels of sperm DNA fragmentation and an increase in the percentage of motile sperm were observed in men aged >40 and in men with BMI > 25. Further, these parameters were significantly improved in the dietary subgroup defined by a high level of coffee intake. Total cholesterol and low-density lipoprotein-cholesterol levels decreased significantly in men <40 years after Aronia supplementation. No statistically significant effects were observed regarding blood pressure, markers of blood sugar regulation, hemoglobin A1c, superoxide dismutase, catalase, isoprostane levels, high sensitivity C reactive protein, or other semen parameters. Conclusions: This study demonstrated a significant increase in glutathione levels and improvement of cytoprotective targets following Aronia supplementation in specific subgroups of men >40 years of age and BMI > 25 but did not demonstrate a significant effect in the overall analysis. The observed concurrent increase in glutathione levels and improvement of cytoprotective targets following Aronia supplementation in subgroups of men, suggests that the endogenous phase II antioxidant glutathione is involved in the modulation of the observed cytoprotective effects. This study is a good foundation for further investigation of these cytoprotective effects in groups with oxidative stress in a dose–response study.

## 1. Introduction

Berries of *Aronia* spp. are known to induce both direct and indirect antioxidant effects [1,2] This capacity has been attributed to the high levels of polyphenols, including anthocyanins [3,4,5,6,7,8]. Aronia intake, either as berries or as juice, has been demonstrated to have beneficial effects on a range of health targets related to oxidative stress (OS), including modulation of OS [6,7,8,9,10,11,12,13], DNA damage [14], inflammation [1,13], semen quality [15], and cardiovascular risk factors, i.e., blood glucose levels [16], endothelial function, obesity, lipid accumulation [17,18], and blood pressure [10,19,20].

OS is an imbalance that occurs when the production of reactive oxidative species (ROS) exceeds the natural antioxidative defense system of the body, causing cellular damage [3,21,22,23,24]. OS has been related to both dyslipidemia, atherosclerosis, hypertension, and diabetes, which are well known risk factors for cardiovascular diseases (CVDs) [4,6,23,25,26,27,28,29]. OS has also been related to sperm DNA instability [21,30,31] expressed by a DNA Fragmentation Index (DFI) [32]. Although the clinical value of DFI is controversial, increased DFI has further been linked to decreased sperm count and motility [33]. It has recently been observed that sperm is a better biomarker of DNA damage than somatic cells such as lymphocytes as mature sperm are unable to repair DNA damage [34]. In this study, we included assessment of sperm DNA as an in vivo model for the determination of the cytoprotective effects of supplements containing bioactive compounds. The above mentioned elements are highly relevant, since both dyslipidemia [35], increased mortality due to CVDs [36], and a high frequency of poor semen quality in Danish men [37] are topics of a major concern.

An important effect of OS is the ROS induced signaling mechanisms that are involved in inflammation and tissue injury [38]. The pleiotropic anti-inflammatory effects of polyphenols may be exerted through regulation of inflammatory cellular activities [23,39,40,41], activation of phase II enzymes [3,42,43], and free radical scavenging limiting the inflammatory response [3,39,44,45]. To counteract OS caused by ROS, the phase II group of antioxidant enzymes, including superoxide dismutase (SOD), catalase (CAT) and glutathione S-transferase may be activated. Anthocyanins have been shown to modulate the activity of phase II enzymes and they may therefore modulate OS in an indirect manner [46].

The main objective of this prospective, double-blinded, randomized, crossover study has been to investigate the influence of Aronia intake on the activity of three phase II enzymes and simultaneously detect potential cytoprotective effects by measuring isoprostane formation in the blood, sperm DNA fragmentation, and sperm quality. Further, we investigated whether Aronia supplementation influences other major health targets associated with OS, including inflammation, blood lipids, markers of blood sugar regulation, and blood pressure. As lifestyle and age are factors that are known to influence exposure to OS, participants were grouped according to age, BMI, and other lifestyle factors.

The investigations of health effects of Aronia published so far have been carried out administrating products based on the cultivated big fruited *Aronia* hybrid × *Sorbaronia mitschurinii*. The health effects of the non-hybrid wild Aronia species *(Aronia melanocarpa, A. prunifolia* and *A. arbutifolia*) included in this study, have not been studied previously although these species have a composition of polyphenols significantly distinct from that of × *Sorbaronia mitschurinii* [47,48].

## 2. Materials and Methods

### 2.1. Study Design

The double-blinded, crossover, randomized controlled trial (RCT) was performed from October 2017 to March 2020 at The Regional Hospital in Horsens, Denmark. The periods of Aronia and placebo supplementations and the wash-out period lasted 90 days each, here considering a full cycle of sperm cell maturation (72–74 days) [49] (Figure 1).

### 2.2. Recruitment and Randomization of the Study Population

Participants were recruited continuously via advertisement in local newspapers, magazines and on social media. All study participants must after oral and written information on the study sign a consent form prior to participation and randomisation. The study including the research biobank has been approved by the Ethics Committee of Region South Denmark (S-20160128). Conduct of the study has been notified and approved by the Data Protection Agency (no. 1-16-02-767-17).

Inclusion criteria were healthy men aged 18 years or older with moderately elevated total cholesterol (TC) levels (5.0–7.0 mM). Exclusion criteria were elevated cholesterol levels due to genetic predisposition, current treatment for elevated cholesterol levels, use of medications that might influence cholesterol levels (including thiazides), genital abnormalities, previous or ongoing handling of heavy metals or toxic substances, and severe disease, e.g., previous cancer or transient cerebral ischemia. Men with high TC levels were not included in this study due to a need for more intensive care. Men with unusual dietary compositions (e.g., alcoholism), consumption of strong vitamins three months prior to enrolment, and mineral or antioxidant dietary supplementations (regular multivitamins were not considered as such) were also excluded from this study. Vasectomy was not an exclusion criterion since sperm parameters were a secondary outcome in this study.

The sample size was based on a with-in standard deviation of HDL of 11.5 and assuming an effect size of 5.3. A sample size of 100 is sufficient to obtain a power of 90 using a one-sample *t*-test. The randomization was performed as block-randomization with block size 10 by an independent statistician using the statistical package Stata (Figure 2). All participants, laboratory technicians, statisticians, and medical personnel were blinded to the randomization. The blinding was first revealed when data had been statistically analyzed from a predefined frame.

### 2.3. Intervention

The daily intake of Aronia supplementation consisted of 5 g freeze-dried seed-free preparations milled to a fine powder, corresponding to the DW of of the pulp from 25–35 g of fresh berries. The capsulated Aronia composition (ARO3:5^®^, Byrial ApS, DK) was standardized according to the content of a total anthocyanins of 150 mg daily and comprised the three wild *Aronia* spp. (*A. arbutifolia* var. Virinia (9%), *A. prunifolia* var. Vikilia (9%), and *A. melanocarpa* var. Swecia (3%)) and the cultivated Aronia hybrid × *Sorbaronia mitschurinii* var *Roar* (79%). This dosage of Aronia anthocyanins was divided into nine capsules to be consumed three times a day (3 capsules with breakfast, 3 at lunch and 3 with dinner). The capsules (Vcaps Plus, Hypromellose, Capsugel, *F. opague* dark green color V18.904) were filled and packed in containers with 810 capsules by Legosan, S. The concentration of anthocyanins in the capsules was determined by liquid chromatography–diode array detection (DAD)–mass spectrometry (MS) according to the procedure described by Wu et al. [51] with some modifications. Briefly, Aronia material from capsules (2 g) was extracted with 50 mL ethanol-water-formic acid (70:27:3). The extraction mixture was sonicated for 20 min under magnetic stirring for 20 min at room temperature, centrifuged for 15 min followed by filtration of supernatant (0.45 µm Kinesis KX Syringe Filter, PTFE filter) and analyzed for anthocyanins by LC-DAD-MS without any dilution. LC-MS data were generated on a LTQ XL (2D Linear Quadrupole Ion Trap, Thermo Scientific, Waltham, MA, USA) mass spectrometer operated in atmospheric pressure chemical ionization (APCI) in negative mode and attached to an Accela HPLC pump and a DAD. Anthocyanins were separated on a LiChrosorb^®^ reversed phase C18 column (5 μm; 250 mm × 4.6 mm, Merck Millipore A/S, Hellerup, Denmark). The mobile phase consisted of aqueous 1% formic acid (solvent A) and methanol (solvent B). The following gradient was used: 20% B at 0 min; 68% B at 30 min; 100% B at 40–43 min; 20% at 45–50 min. Column temperature: 30 °C, flow rate: 0.8 mL/min, and injection volume: 10 μL. UV detection: 520 nm. The major anthocyanins cyanidin-3-galactoside (*t*_R_ = 15.8 min) and cyanidin-3-arabinoside (*t*_R_ = 17.9 min) were quantified by external calibration curves.

The placebo capsules were identical to the Aronia preparations in color and contained microcrystalline cellulose (same weight as Aronia capsules, Legosan, S.). To clarify compliance to suppl. intake, the remaining capsules were accounted for at the regular visits.

### 2.4. Clinical Tests and Questionnaires

Prior to and after the intervention periods, two blood samples and two semen samples were provided from each participant, 4–8 days apart (Figure 1). Blood pressure was measured at each consultation after 15 min of rest. Semen samples were delivered by masturbation, mostly at home, kept at body temperature, and delivered to the Regional Hospital, Horsens, within two hours. Only a few samples were made at the hospital. Considering standardization, all tests were collected in the morning, and the participants were fasting.

Concomitantly with enrollment, participants were asked to fill out a questionnaire on lifestyle habits, chronic or hereditary illnesses, and previous ability to father a child, among others. In addition, the participants were twice asked to fill out a questionnaire regarding side effects after the first and second period.

### 2.5. Analytical Procedures, Outcomes and Statistical Methods

#### 2.5.1. Blood Tests

The primary outcomes of blood samples were markers of OS and inflammation, serum lipid levels, long-term blood glucose (HbA1c), homeostatic model assessment (HOMA), and fasting glucose. The blood samples were analyzed in standardized hospital laboratories.

To investigate lipid status, TC, triglycerides (TG), low-density lipoprotein cholesterol (LDL-C), and high-density lipoprotein cholesterol (HDL-C) were analyzed.

C-reactive Protein (CRP) was the primary marker of inflammation. To analyze CRP-counts, the high sensitivity CRP (hs CRP) was used, as this test can detect slight changes within the normal ranges of CRP [52].

Conditions of OS were measured by levels of glutathione, SOD, CAT, and isoprostanes. Glutathione, SOD, and CAT are part of the antioxidant defense system scavenging ROS [30,53,54], and isoprostanes are regarded as valid indicators of cell membrane damage caused by OS [55,56].

#### 2.5.2. SOD Determination

In order to determine the SOD activity, a SOD Determination Kit was used (Sigma Aldrich, St. Louis, MO, USA, 19160). All standards and samples were done in triplicate and measurement were taken using a UV spectrophotometer plate reader (Thermo Scientific Varioskan Lux).

#### 2.5.3. Glutathione Assay

Glutathione was measured using a Glutathione Assay kit (Sigma Aldrich, CS0260). The plasma sample was treated with 5% 5-Sulfosalicylic Acid and centrifuged at 10,000× *g* for 10 min. Reduced glutathione (GSH) was measured on a UV spectrophotometer plate reader (Thermo Scientific Varioskan Lux) at 412 nm.

#### 2.5.4. Catalase Activity Assay

Catalase was measured using a Catalase Activity Colorimetric Assay Kit (BioVision K773-100, Abcam Assay, Cambridge, UK). A catalase positive control, a high control and blood plasma samples were measured. Measurements were done using a UV spectrophotometer plate reader (BioTek Synergy H1) at 412 nm.

#### 2.5.5. Isoprostanes

Concentrations of 8-iso-PGF in human plasma samples were determined by the utilization of a Competitive Enzyme-linked Immunosorbent Assay kit (Enzo Life Sciences, AD900-091). The plasma samples were treated with 10 M NaOH (VWR) and subsequently with 12.1 M HCl (VWR). All samples were pH adjusted and analyzed in triplicate. The measurements were conducted on a UV spectrophotometer plate reader (Epoch microplate spectrophotometer) with Gen5 microplate data software at 405 nm.

#### 2.5.6. Insulin and Glucose

Plasma insulin was measured by ELISA technique using commercial kits (Agilent Dako Denmark A/S, K2379) with intra-/inter-assay precision of 5.1–7.5% and 4.2–9.3%. Plasma glucose was measured on Cobas c111-system by standard enzymatic colorimetric assays using commercial kits (Roche Diagnostics Gmbh, 04657527). Intra-/inter-assay precision were between 0.8–1.1% and 0.5–0.6%.

#### 2.5.7. Handling of Semen Samples

Semen samples were handled according to WHO guidelines criteria [57], except that not all were analyzed within one hour from ejaculation as WHO guidelines required (the maximum time gap from ejaculation to analysis was 3 h considering one sample), as most semen samples were made at home. However, each participant was individually consistent concerning place of ejaculation and time to deliverance of semen samples during the study period. Liquefaction was completed and followed by a determination of sperm concentration, count, motility, and morphology. This was estimated through investigation by manually counting in duplicate, using a Makler Counting Chamber through a light microscope with phase-contrast at 200× magnification. All counts and motility scores were performed by the same experienced medical laboratory technician.

Semen quality was determined by total sperm concentration and motility, defined by total motile sperm count (TMSC) and total progressive motile sperm count (TPMSC). The concentration of motile sperm (CMS) and concentration of progressive motile sperm (CPMS) were also estimated. Additionally, sperm DNA-integrity was assessed with the SDI^®^-test (SPZ Lab A/S, Copenhagen, Denmark). This test is based on the staining protocol for the SCSA^®^ [58] but with a more detailed quality control in the flow. The DNA Fragmentation Index (DFI) was performed on two replicates for each semen sample.

#### 2.5.8. Compliance

A participant was considered compliant to the clinical tests if one out of two measurements of the baseline and follow up periods, in the given parameter, was collected. This resulted in at least four measurements with one attendance in each period.

#### 2.5.9. Primary and Secondary Outcomes

In the protocol the results of this study were originally planned to be published in three publications: 1. A hypothesis generating study on the effects on oxidative stress. Outcomes: SOD, CAT, GLU, isoprostanes and Hs CRP. 2. Confirmatory study on effects on serum-cholesterol levels with primary outcomes, TC, LDL-C, HDL-C, TG, secondary outcomes: Blood pressure, SOD, CAT, GLU, isoprostanes, HbA1c, DFI and Hs CRP. 3. Confirmatory study on effects on sperm quality with primary outcomes: TMSC, TPMSC. Secondary outcomes: Pregnancy during the study or within 3 months of terminating participation in the study, SOD, CAT, HbA1c, DFI and Hs CRP, testosterone, blood pressure. During evaluation of the results of the clinical study it was evaluated to be more sensible to publish all results in one paper presenting the primary and secondary outcomes listed above.

#### 2.5.10. Statistical Analyses

##### Crossover Analyses

In the crossover statistical analyses, baseline data were not included, and solely differences in mean follow-up data (outcome) were compared. For continuous outcome, the treatment effect was defined as mean outcome for Aronia (A) minus mean outcome for placebo (P). An average measurement was calculated when two measurements in each period were available. Treatment effect was estimated using the two-sample unequal variance *t*-test as the mean of the AP combination minus the PA combination [59].

A secondary crossover analysis was performed for glutathione using baseline measurements, where the treatment effect of Aronia compared to placebo in each of the AP and PA groups has estimated using a paired *t*-test.

Baseline characteristics were expressed as mean ± SD for continuous data and *n* (%) for binary data. Statistical significance was set at *p* < 0.05.

##### Subgroup Analyses

The Aronia effect in subgroups was estimated since randomization among men in the subgroup ensures equal distribution of known and unknown risk factors in the Aronia and placebo group, at least if the subgroup is large enough. Upon recommendation, no adjustments for observed differences in baseline risk factors were made [60].

All data were divided into subgroups regarding BMI (BMI ≤ 25 and BMI > 25 kg/m^2^), age (<40 and ≥40 years of age), smoking (smokers and non-smokers), lifestyle and dietary habits, and glutathione levels at baseline 1.

The limit separating BMI subgroups was set at 25 kg/m^2^, as individuals with a BMI > 25 kg/m^2^ are considered overweight [61]. Furthermore, studies investigating the effect of BMI on male fertility found a dose–response relation from BMI > 25 kg/m^2^ [62]. The age limit at 40 years was decided, since DFI levels increase with age [63].

##### Test for Carry Over-Effect

Whether a carry over-effect of the Aronia supplementation was present was investigated by calculating the sum of the two outcome measurements for each patient and comparing the two groups by using the two-sample unequal variance *t*-test [64]. The baseline levels in the two groups were assumed to be similar due to the randomization, and the baseline levels in group A and B were calculated and compared. Test for carry over-effect was solely conducted on data regarding glutathione.

## 3. Results

In the following, data on participant characteristics and markers of cytoprotection, including glutathione, DNA fragmentation, percentage of motile sperm, and isoprostanes will be presented together with effects on serum lipids. For detailed results on markers of blood sugar regulation, HbA1c, blood pressure, SOD, CAT, hs CRP, and remaining sperm parameters (TPMSC, CMS, CPMS), see the Appendix A.

### 3.1. Participant Characteristics

Of the 109 eligible participants assigned, 95 completed the study. At baseline, participants in group A and B did not differ significantly in most of the parameters except for a significant difference in age (*p* = 0.03) (Table 1).

The non-compliant group consisted of 14 study participants, leading to a total compliance of 87.2% regarding blood samples. This group did not significantly differ in baseline demographics compared to the compliant group except from the distribution of smokers. In the non-compliant group, 4 out of 14 participants (28.6%) were smokers, whereas 4 out of 95 participants (4.2%) in the compliant group were smokers.

Thirty-four out of 109 participants were vasectomized at baseline, and an additional two participants were vasectomized during the trial. For two participants, who were considered compliant to the study, semen analyses on all samples failed due to inhomogeneity after a minimum of 35–60 min. One participant was excluded, as no motile sperm in his semen samples were detected. These participants were not categorized as non-compliant but were excluded in further calculations concerning semen parameters. A total of 8 men out of the remaining 70 were non-compliant for unknown reasons, resulting in a total compliance of 88.6%. The 47 excluded men did not differ according to baseline demographics.

The first participant was included in the study February 2018 and the last participant finished the second period of administration in January 2020. Inclusion was terminated as planned after 109 participants had enrolled.

### 3.2. Markers of Cytoprotection

#### 3.2.1. Glutathione

An overall significant increase in glutathione levels after Aronia intake compared to placebo was observed (+34.9 mM, *p* = 0.0095) when baseline data were included in the statistical analysis and +19.8 mM, *p* = 0.15 when excluding baseline data (Table 2). In the subgroup of men with low glutathione levels (<42 mM) at baseline 1, glutathione was increased significantly after intake of Aronia compared to placebo tablets (+41.3 mM, *p* = 0.038; Table 2). Further, subgroup analyses on study participants with a high coffee intake (3 cups or more daily) showed statistically significant increased glutathione levels after Aronia supplementation (+32.6 mM, *p* = 0.045; Table 2).

No significant effects of Aronia supplementation were found in the remaining subgroups with respect to glutathione. Besides a positive effect in the Aronia intake period, graphic data indicated a potential long-term effect of Aronia, since baseline 2, after the 90 days of wash out only decreases halfway back to the original baseline 1 level (Figure 3). However, statistical tests showed no significant carry over-effect of the Aronia period (*p* = 0.71).

#### 3.2.2. Isoprostanes

Although both the overall and most subgroup crossover analysis comparing follow-up measurements after Aronia and placebo supplementations showed lower calculated isoprostane levels after Aronia supplementation (all participants, −46.7 pg/m (*p* = 0.83) and subgroup BMI > 25, −106.8 pg/m (*p* = 0.73)) these differences in means were not statistically significant. 

### 3.3. Sperm Quality

#### 3.3.1. Sperm DNA Fragmentation

Statistically significant decreases in DFI were found in the subgroups BMI > 25 kg/m^2^ (−2.4, *p* = 0.037) and age ≥ 40 years (−2.0, *p* = 0.032) after supplementation with Aronia (Table 2). An overlap between these two subgroups was detected, as 81.1% of the participants ≥40 years of age also had a BMI > 25 kg/m^2^. Statistically significant decreases in DFI after Aronia administration were also found in subgroups regarding dietary habits (Table 2). This effect was observed in the following subgroups: no vitamin D supplementation (−1.8, *p* = 0.043), no fish oil supplementation (−2.1, *p* = 0.031), high fruit and vegetables intake (−5.6, *p* = 0.031), low daily sugar intake (−2.7 *p* = 0.032), sitting down < 6 h daily (−2.3, *p* = 0.046), low alcohol intake (−1.2, *p* = 0.037), and high coffee intake (−3.2, *p* = 0.01). The statistical analysis based on all participants did not show a significant decrease in DFI levels in the Aronia groups although the level of DFI was 1.2 percentage units lower in the Aronia treated group (*p* = 0.23; Table 2) compared to placebo groups.

#### 3.3.2. Percentage of Motile Sperm (PM)

Significant increases were found in the subgroup analysis regarding effect of Aronia supplementation on PM in the groups of men aged ≥40 years (+5.4, *p* =0.004), BMI > 25 (+3.6, *p* = 0.052), non-smokers (+3.4, *p* = 0.036), low alcohol intake (+4.6, *p* = 0.01), daily intake of 3 or more cups of coffee (+5.7, *p* = 0.006), no magnesium supplementation (+3.6, *p* = 0.024), and no calcium supplementation (+3.6, *p* = 0.024) (Table 2). The crossover analysis including all participants showed an increase in the calculated mean PM after Aronia administration (+2.7, *p* = 0.067), although this was not statistically significant.

### 3.4. Serum Lipids

#### 3.4.1. Total Cholesterol

The subgroup analysis including men <40 years found a statistically significant decrease in TC levels after Aronia supplementation compared to placebo supplementation (−0.4, *p* = 0.046; Table 3). In the subgroup regarding men drinking 2 or less cups of coffee per day, the mean TC level decreased after the Aronia period (−0.2, *p* = 0.052; Table 3). The crossover analysis including all men showed lower calculated mean TC levels after supplementation with Aronia, but this difference was not statistically significant (−0.1, *p* = 0.16; Table 3).

#### 3.4.2. LDL-Cholesterol

Subgroup analysis on men drinking two or less cups of coffee per day showed a significant decrease in LDL-C when treated with Aronia (−0.2, *p* = 0.033; Table 3). Crossover analysis including all men showed lower, but non-significant, mean LDL levels following Aronia supplementation (−0.1, *p* = 0.43).

#### 3.4.3. HDL-Cholesterol and Triglycerides (TG)

No statistically significant effects regarding levels of HDL-C and TG were found in the crossover and subgroup analyses (Table 3).

### 3.5. Side Effects

No difference in the reporting of side effects were registered among group A and group B, respectively. Side effects was reported by 20 participants (group A: *n* = 12; group B: *n* = 8) after the first treatment period and by 15 participants (group A: *n* = 11; group B *n* = 4) after the second treatment period. Of these participants, 8 reported having experienced side effects after both periods, resulting in a total of 27 participants (24.7%) reporting side effects (Table 3). Eight participants solely reported side effects after receiving the Aronia suppl., and 11 participants solely reported side effects after receiving the placebo suppl. The mentioned effects were primarily gastrointestinal discomfort and secondly fatigue; 22.9% of participants in total reported having experienced one or several gastrointestinal side effects during one or both treatment periods, and 6.4% reported fatigue. Fifteen participants reported on daily or almost daily experiences of side effects. No study participants left the study due to side effects.

## 4. Discussion

### 4.1. Effects of Aronia on Cytoprotective Targets

The fundamental idea behind the design of this clinical trial was to screen a group of men for potential health effects induced by a daily intake of a realistic amount of Aronia over a 90-day period. This was performed to select targets for further detailed studies on a more homogeneous group of participants, e.g., levels of OS, glutathione or DFI as inclusion parameter. The present study clearly points out well defined targets that are positively influenced by Aronia supplementation. It is notable that most of these targets are indicators for cytoprotective effects, i.e., glutathione in blood serum, DFI and the PM. Further, the data on isoprostanes likewise may suggest that cell membrane damage is lowered by Aronia intake although the observed decrease in mean isoprostane levels were not statistically significant.

The results in the present study, demonstrating that Aronia intake significantly increases glutathione levels after 90 days administration together with the concurrent improvement of DFI and PM in men aged >40 and/or have a BMI > 25, have not been shown previously. It is, in our opinion, important knowledge to understand the physiological processes explaining the positive health effects on a range of health targets following Aronia intake reported in a range of clinical trials [4,6,11,19,42,65,66,67,68,69,70].

### 4.2. Glutathione

The significant increase of glutathione in men that had a relatively low glutathione level at first baseline indicate that it is possible to screen men for low glutathione levels and subsequently raise the level of this endogenous antioxidant to counter OS. The results of this study indicate that this intervention may improve the DFI and PM.

### 4.3. DNA Fragmentation and Percentage of Motile Sperm

The observation that the lowering of DFI and the increase in PM were most positively affected in men aged >40 and/or BMI > 25, is in accordance with other clinical studies regarding the connection between obesity and decreased male fertility [71,72,73,74]. Reduced fertility in obese men may be due to an increase in estrogens and a decrease in androgens [62,74]. A study by Kort et al. [73] found a significant difference in DFI in overweight and obese men when compared to men with normal BMI. This indicates that obesity may be associated with sperm DNA damage, or a reduction in DNA-repair/defense mechanisms. However, the study by Kort et al. [73] did not address OS as a contributing factor and more studies should be made to clarify a possible connection between OS and sperm DNA damage and/or DNA repair mechanisms.

In the subgroups regarding men aged >40 years and BMI > 25, DFI decreased concomitantly with an increase in motility when the supplementation was Aronia. Since DFI and motility may be affected by OS [75], originating from leukocytes and dead sperm [24], anthocyanins and/or other polyphenols from Aronia might play a role in decreasing semen DFI levels in obese men and in men aged 40 or above, suggesting a cytoprotective effect. The fact that a significant subgroup of the men included in this study were vasectomized weakened the statistical analysis concerning sperm parameters. Another problem was that more older men participated in the trial and these may have had longer abstinence times than the younger men [63]. Hence, more studies are needed to investigate if OS in semen is increased in obese men, and whether antioxidant supplements might prevent damage to sperm.

The availability of high-quality clinical trials investigating the effect of Aronia extract on human sperm is scarce, thus emphasizing the importance of this study. Studies investigating the effect of polyphenols, and other antioxidants, on human sperm have reported favorable outcomes concerning motility and sperm count [76,77,78], which is in accordance with the results regarding DFI and TMSC in our study. A study by SiIberstein et al. [77] showed that men with greater antioxidative capacity in their semen had better sperm count and motility parameters. The study suggested that an indirect effect of polyphenols might complement the antioxidative capacity within the semen [77]. Our study did not assess the antioxidative capacity within semen but used DFI levels to evaluate the effect of OS on sperm.

Several clinical studies on male infertility included men with reduced semen quality [75,76,78]. The meta-analysis by Santi et al. [79] determined a DFI of 20 as a suitable cut-off value, when determining male fertility. Another study by Vaughan et al. [75] investigated the inter-relationships between OS, DFI and age, and found that both DFI and OS increased with age and were associated with defects in the spermatogenesis. The study categorized DFI into 3 groups: low (<20), medium (20–30) and high (>30). In our study, 60% of the participants had a seminal DFI below 20 before intervention, and the mean baseline DFI value was 20.2, which is near the lower cut-off value suggested by several authors [75,79,80]. This puts the participants of our study in the category with low sperm DFI according to both above mentioned studies. Our study population consisted of men older (mean age 50.7) than the men in the study by Vaughan et al. (mean age 37.6). The lower DFI and older age in our study does not entirely correspond to their results. Nonetheless, our study did find significant results of Aronia in the subgroup regarding age ≥40 years, suggesting a higher amount of OS and DFI in this category, indicating the same tendency discovered in the above-mentioned studies [75,76,78]. If our study population only comprised men with low semen quality, effects that are more significant might have been achieved on several parameters.

### 4.4. Significant Cytoprotective Effects of Aronia among Coffee Drinkers

The clear positive effect on both glutathione levels, DFI, and PM after Aronia consumption observed in the subgroup drinking 3 or more cups of coffee a day is notable. A review by Ricci et al. [81] concluded that caffeine intake may negatively affect male reproductive function, possibly through sperm DNA damage.

Glutathione levels may, however, in fact be raised by moderate coffee drinking [82]. Hence, a lower glutathione level, a higher DNA damage, and a lower motile sperm percentage at baseline 1 in the high coffee intake group would explain the observed significant effects, but in our study, no difference at baseline regarding glutathione levels, DFI or PM in subgroups were found at the time of initiation of the trial (data not shown). Therefore, it is suggested that the observed positive effect of coffee drinking together with Aronia intake may be explained by a yet unknown factor increasing the bioavailability of anthocyanins and other polyphenols from Aronia. Studies that may support this hypothesis include the study of Takahashia et al. [83] demonstrating that intestinal absorption of the anthocyanin, cyanidin-3-glycosides from Aronia, is increased after intake of capsaicin and capsiate from chili in a rat ligated small intestinal loop model and Kim et al. [84] reporting that coffee bean extract increases the absorption of the phenolic compound salicylic acid in mice.

### 4.5. Effects of Aronia on Blood Lipid Levels

The statistically significant but moderately positive effect of Aronia administration on TC and LDL-C observed in the subgroup analyses suggests that higher doses of Aronia are necessary to induce more significant decreases in blood lipid levels, or that Aronia supplementation potentially has a more significant effect on lowering TC and LDL-C in persons with moderate or severe hypercholesterolemia. This is supported by the clinical studies of Skoczynska et al. [65] who investigated the effects of oral Aronia consumption during 6 weeks on 58 healthy, mildly hypercholesterolemic men and found beneficial effects on blood lipids and serum glucose levels after daily intake of Aronia containing >500 mg anthocyanins. These findings could potentially be due to the rather high concentration of daily anthocyanin consumption, compared to the concentration in our study (150 mg anthocyanin/day). A study by Duchnowics et al. [67] on 25 healthy participants with hypercholesterolemia found decreases in cholesterol levels and lipid peroxidase after two months of daily Aronia supplementation containing approximately 60 mg of anthocyanins. The effects on cholesterol levels in the latter study, compared to our study, could be explained by the inclusion criteria of high levels of TC (>13 mM) among study participants, compared to TC levels of our study participants (5–7 mM).

The effects of Aronia supplementation on markers of OS and CVD risk factors found in our study is supported by several clinical trials investigating the effects of Aronia supplementation on different patient groups. Studies conducted on patients with mild hypercholesterolemia [65,67], metabolic syndrome [6,19], and after myocardial infarction [66] found significant treatment effects on markers of OS and significant decreases in CVD risk factors, e.g., blood lipids and serum-glucose after daily intake of anthocyanins [4,6,19,66,85,86]. Other RCTs performed on healthy patients have found improvements of antioxidant status and beneficial effects on blood lipids after daily consumption of anthocyanins [68,69,87,88]. These findings may be due to higher doses of anthocyanins, younger study populations, and short intervention periods [69,87,88] compared to our study. The study group in our study primarily consisted of middle aged, non-smoking, fertile, mildly hypercholesterolemic men. The good health of the participants may lead to a less distinct effect of Aronia consumption on the above-mentioned parameters compared to other studies, with study populations composed of patients with chronic diseases [4,6,19,65,67,85]. This might have contributed to the non-significant results in the general crossover analyses including all men regardless of age, BMI, glutathione level, etc. at baseline and could potentially explain why statistically significant results were solely found in subgroup analyses.

### 4.6. Side Effects

In this study, potential side effects of Aronia supplementation were investigated. Twenty-seven participants (24.7%) reported side effects after one or both treatment periods. The complaints were primarily different variations of gastrointestinal discomfort. Side effects were in many cases reported after both treatment periods, and whether these reported effects were caused by the actual Aronia preparations or were due to the placebo capsules was not determined. The importance of the reported side effects is considered of less concern as no study participants left the study due to side effects and since side effects were more frequently reported in the placebo group compared to the Aronia group.

### 4.7. Strengths

Several reviews have compared studies investigating dietary antioxidative supplementation, and nearly all suggest better designed placebo-controlled studies to investigate the physiological benefits of Aronia consumption [8,16,21,32,89,90].

The solid study design was a strength of this study. The prospective, double-blinded, crossover RCT design eliminated possible confounders and differences in baseline demographics and counterbalanced the interindividual treatment effects of Aronia.

The overall compliance of 87.2% for blood samples and 88.6% for semen samples was a strength. Since the non-compliant participants did not differ in baseline demographics, except for the predominance of smokers, the influence of the treatment effect on this drop-out was of less concern.

The size of the study population with more than 100 men enrolled and high compliance was an outstanding strength of this study. This strengthened the representativeness of the study population compared to the general population and made the study one of the largest RCT studies investigating Aronia supplementation. Furthermore, the long study period of 270 days created a better opportunity for treatment effects to be observed. The wash-out period of 90 days minimized the risk of a carry over-effect, which could have influenced the results of the second treatment period. In this study, a test for carry over-effect was made on glutathione, as there was indication of long-term effects of Aronia on this parameter (Figure 3). The result did not show any significant carry over-effect of Aronia.

The only significant differences among group A and B in baseline demographics were age and blood pressure. Whether Aronia had a potential carry over-effect in specific age groups remain unclear. The blood pressures in group A and B showed small interindividual variation within the normal reference range and were therefore of less importance. Furthermore, the mean baseline levels of glutathione in the two groups were compared and showed no significant differences, strengthening the test of a potential carry over-effect.

### 4.8. Limitations

The good health of the study population might have camouflaged even small changes and positive effects of Aronia. The subgroup analyses were not adjusted for confounding, since randomization among men in the subgroups ensures equal distribution of known and unknown risk factors in the Aronia and placebo group. As mentioned previously, an overlap between the subgroups BMI > 25 kg/m^2^ and age ≥ 40 years old was detected. Regarding this overlap, it is not currently possible to conclude if the change in effects is due to BMI or age. To address this potential issue, it is necessary to investigate Aronia’s effect in combination of BMI > 25, BMI ≤ 25 and age > 40, age ≤ 40 years, for example in a regression analysis, but such an analysis would require a larger sample.

The daily dose of dried Aronia powder corresponded to a total of 150 mg anthocyanins per day, which was considered to be a realistic dose of daily antioxidative supplementation intake. A systematic review of the clinical trials on Aronia products concluded that daily doses of Aronia preparations should contain between 300 and 600 mg anthocyanins to achieve significant results [8]. Given results achieved by a far lower dose of anthocyanins [67,68,70] in other clinical trials and in consideration of achieving a satisfying compliance, the dose of 150 mg anthocyanins was chosen for this study. In addition, when deciding on the daily dose of anthocyanin intake, the risk of adverse side effects caused by high doses and the risk of low compliance to suppl. intake, was considered.

The included study participants were diagnosed with mild hypercholesterolemia and therefore might have been highly motivated to lower their cholesterol levels, consequently altering their lifestyle during the study period, and thereby masking potential beneficial effects of Aronia.

Nearly all semen samples were made at home, kept at body temperature, and transported to the clinic for analysis. More than half (64.9%) were analyzed >1 h from ejaculation, which is not in accordance with the WHO 2010 guidelines [57]. This might have led to an underestimation of the treatment effect of Aronia on semen parameters in this study. However, since most men maintained continuity in deliverance of their semen samples throughout the study, any longitudinal changes present would most likely have been detected.

## 5. Conclusions

This prospective, double-blinded, randomized, crossover study on mildly hypercholesterolemic men did not demonstrate a significant effect of the Aronia supplements in the overall analysis. However, in the subgroup analysis on men > 40 years of age, BMI > 25 and dietary supplement groups such as a high intake of coffee, supplementation with relatively low levels of the Aronia preparation induced statistically significant positive effects on cytoprotective targets; DNA fragmentation and PM. The observed concurrent increase in glutathione levels following Aronia intake suggests that this endogenous phase II antioxidant is involved in the observed cytoprotective effects. The study showed that it is possible to raise the glutathione levels in men significantly with low baseline glutathione status by the tested Aronia supplementation. Further, data indicate that this positive effect on glutathione levels may last beyond the treatment period. The relatively low daily dose of anthocyanins and the rather healthy study population, compared to other studies, may explain the relatively limited positive effect on cholesterol level and the non-significant results on a range of targets in the general crossover analyses including all men across subgroups. The results from the present study form a good foundation for further investigation of the cytoprotective effects in groups with OS in a dose–response study. This may also target the observed significant positive effect of Aronia in participants with a high daily coffee intake to investigate if a synergistic effect may be present between Aronia suppl. and coffees content of, e.g., health promoting polyphenols.

## 6. Registration

The study is registered in ClinicalTrials.gov (01 May 2018). The trial registry name is Potential Health Effects of Aronia Intake and the registration number is NCT03405753.

## Figures and Tables

**Figure 1 jcm-12-00373-f001:**
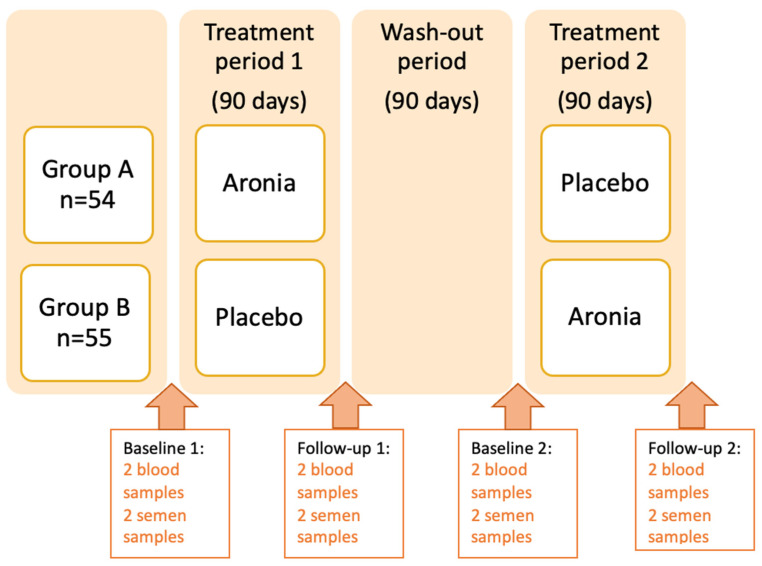
Study Design.

**Figure 2 jcm-12-00373-f002:**
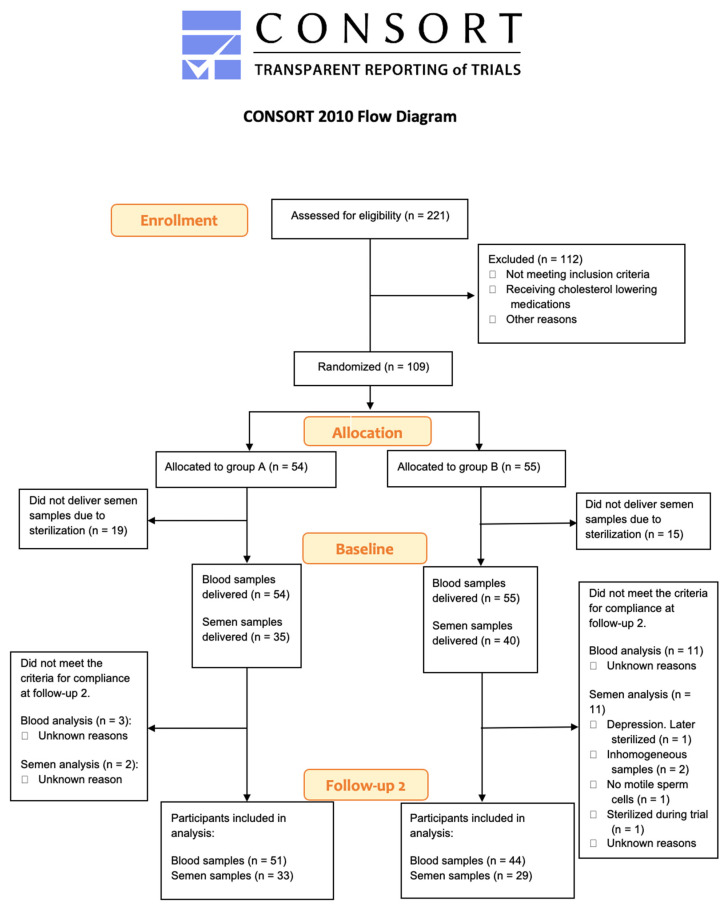
Consort 2010 Flow Chart [50]. Illustrating the flow of the study participants.

**Figure 3 jcm-12-00373-f003:**
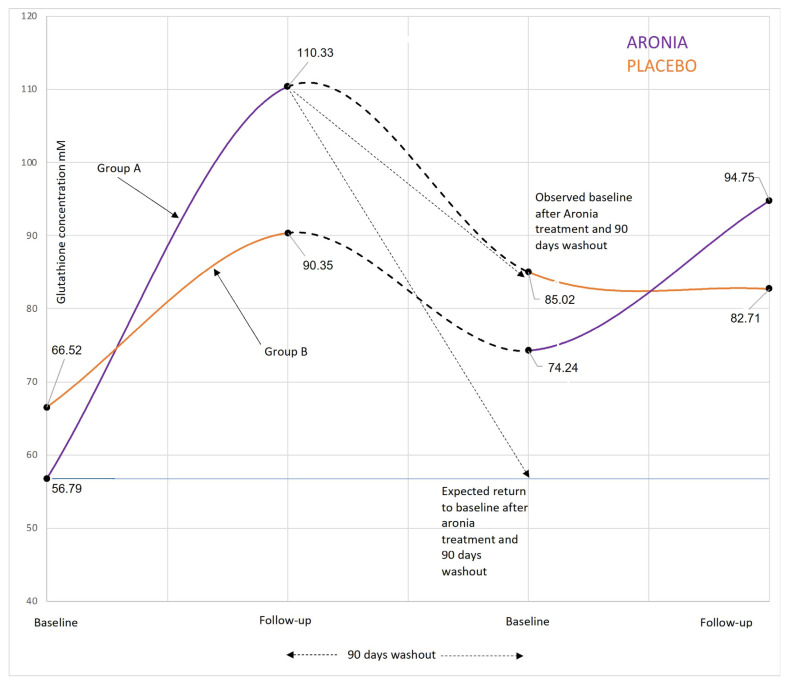
Development in glutathione levels in blood samples following Aronia and placebo supplementation during the 270 days crossover designed. Note the high level of glutathione at follow-up after 90 days of Aronia intake. After the subsequent 90 days wash out, the glutathione baseline would be expected to drop to the level of baseline prior to Aronia supplementation but it only dropped halfway to the original baseline. This indicates a possible long-term effect of Aronia across the washout period. Blue lines indicate periods with 90 days Aronia supplementation, and the orange lines indicate periods with 90 days of placebo administration.

**Table 1 jcm-12-00373-t001:** Demographic data. Baseline values expressed as mean ± SD.

	Group A (*n* = 54)	Group B (*n* = 55)	*p* Value [95% CI]
Age (years)	47.7 ± 9.7	51.9 ± 9.3	0.03 [−7.9; −0.4]
BMI (kg/m^2^)	27.2 ± 3.2	27.3 ± 3.5	0.87 [−1.4; 1.2]
Smoking, *n* (%)	3 (6.1)	5 (9.3)	
Baseline total cholesterol (mM)	5.5 ± 0.8	5.6 ± 0.9	0.73 [−0.4; 0.3]
Baseline blood pressure (systolic) (mmHg)	133.4 ± 14.4	131.9 ± 15.4	0.59 [−4.1; 7.2]
Baseline blood pressure (diastolic) (mmHg)	87.0 ± 10.8	85.3 ± 11.0	0.41 [−2.4; 5.9]
Fasting blood glucose (mM)	5.8 ± 0.4	6.0 ± 1.2	0.32 [−0.5; 0.2]
Hs CRP (mg/L)	2.3 ± 2.3	2.3 ± 3.0	1.00 [−1.0; 1.0]
Glutathione (nM)	56.8 ± 69.9	64.5 ± 83.4	0.60 [−36.9; 21.5]
DFI (%)	19.6 ± 14.0	19.7 ± 12.2	0.98 [−6.3; 6.1]
TPMSC (mio.)	108.9 ± 106.3	85.5 ± 87.7	0.31 [−22.4; 69.2]

**Table 2 jcm-12-00373-t002:** Cytoprotective effects of Aronia supplementation measured on glutathione levels in blood serum, sperm DNA fragmentation, and percentage of motile sperm. Upper part of the table list data on the entire group of men. The lower part of the table illustrates the statistical analysis of subgroups defined according to age, BMI, dietary habits, and level of daily activity. [CI] = Confidence interval.

Analysis of All Participants Incl. Subgroups		Glutathione (mM)		DNA Fragmentation (%)		Motile Sperm (%) (PM)	
Treatment Order		Aronia Then Placebo	Placebo Then Aronia	Aronia Then Placebo	Placebo Then Aronia	Aronia Then Placebo	Placebo Then Aronia
N		51	42	33	27	33	29
Treatment effect [CI]		+19.8 [−7.6;47.1]		−1.2 (−3.3;0.8)		+2.7	
Crossover analysis 1		0.15		0.230		0.067	
Crossover analysis 2, including baseline data		0.0095 **					
**Analysis of subgroups**							
Age > 40	Effect			−2.4		+5.4	
	*p*-value			0.032 *		0.004 **	
	*n*			41		43	
BMI > 25	Effect			−2.0		+3.6	
	*p*-value			0.037 *		0.052	
	*n*			42		43	
Glutathione < 42 mM at baseline 1	Effect	+41.3					
	*p*-value	0.038 *					
	*n*	54					
No D vitamin supplementation	Effect			0.043 *			
	*p*-value			−1.8			
	*n*			51			
No fish oil supplementation	Effect			−2.1			
	*p*-value			0.031 *			
	*n*			49			
High fruit and veggie intake	Effect			−5.6			
	*p*-value			0.031			
	*n*			13			
Low daily sugar intake	Effect			−2.7			
	*p*-value			0.032 *			
	*n*			37			
Sitting down less than 6 h/day	Effect			−2.3			
	*p*-value			0.046 *			
	*n*			40			
Low alcohol intake	Effect			−1.2		+4.6	
	*p*-value			0.037 *		0.01 *	
	*n*			48		50	
No smoking	Effect			−1.9		+3.4	
	*p*-value			0.051		0.036 *	
	*n*			53		54	
High coffee intake	Effect	+32.6		−3.2		+5.7	
	*p*-value	0.045 *		0.01 *		0.006 **	
	*n*	60		37		39	
No magnesium supplementation	Effect					+3.6	
	*p*-value					0.024 *	
	*n*					50	
No calcium supplementation	Effect					+3.6	
	*p*-value					0.024 *	
	*n*					50	

* statistically significant at the *p* ≤ 0.05 level; ** statistically significant at the *p* ≤ 0.01 level.

**Table 3 jcm-12-00373-t003:** Effects of 90 days *Aronia* spp. administration on TC, LDL-C, HDL-C, and TG. CI = Confidence Interval.

Analysis of All Participants Incl. Subgroups		Total Cholesterol (mM)		LDL-Cholesterol (mM)		HDL-Cholesterol (mM)		Triglyceride (mM)	
**Treatment order**		Aronia then Placebo	Placebo then Aronia	Aronia then Placebo	Placebo then Aronia	Aronia then Placebo	Placebo then Aronia	Aronia then Placebo	Placebo then Aronia
** *n* **		51	43	51	43	51	43	51	43
**Treatment effect (CI)**		−0.1 (−0.2; 0.1)		−0.1 (−0.2; 0.1)		−0.0 (−0.1; 0.0)		−0.0 (−0.1; 0.1)	
**Cross over analysis *p*-value**		0.16		0.43		0.12		0.78	
**Analysis of subgroups**									
**Age < 40 years**	*p*-value	0.046 *							
	Effect (CI)	−0.4 (−0.9; 0.0)							
	*n*	15							
**Low daily coffee intake**	*p*-value	0.052		0.033 *					
	Effect (CI)	−0.2 (−0.5; −0.0)		−0.3 (−0.6; 0.0)					
	*n*	28		28					

* statistically significant at the *p* ≤ 0.05 level.

## Data Availability

The data were collected in a RedCap database, and access can be given by the authors.

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
