# Peer review of "Effects of Chokeberries (Aronia spp.) on Cytoprotective and Cardiometabolic Markers and Semen Quality in 109 Mildly Hypercholesterolemic Danish Men: A Prospective, Double-Blinded, Randomized, Crossover Trial"

_jcm, 2023, doi:10.3390/jcm12010373_

Round 1

Reviewer 1 Report

The study is well designed and thought of. It investigates the beneficial cardiometabolic, inflammatory and fertility effects of Aronia on a specific ethnic group of patients (Danish). I would advise to better specify this in the title as "health" deems to generic. The introduction fits its purpose but a figure is missing. Materials and Methods and statistical analysis are described in an orderly fashion however the difference between group A and B regarding age and blood pressure represents a big bias as both these factor are determinants in cardiovascular fertile factors. Results show neatedly interesting evidence however the small sample size does not give satisfaction for a true return. Side effects has a high percentage. The discussion suffers the above bias but Limitations are well described. 

Author Response

Thank you for your constructive criticism.

  1. We have now changed the title so it is more specific. Thus "health markers" has been changed to "cytoprotective and cardiometabolic markers and semen quality". The changes marked with red.
  2. We find it is difficult to include a meaningfull figure in the introduction, and as far as we can see, it is usually not done in other JCM papers. Hope that we have understood you right, and that you can accept that we omit to include a figure in the introduction.
  3. Thank you so much for your good comments to the M&M section. We have now been through all our data, and there were incorrections in Table 1. The correct data shows no significant differences between the blood pressure for Group A and B. Since the old version of Table 1 was uploaded as a picture, we had to exchange it. The new version marked in red.
  4. Best regards
  5. Jens Fedder 

Reviewer 2 Report

The research paper is full of useful details.  The paper could be used as a base for a following research about Aronia spp .  effect on human health.

Author Response

Thank you so much for your positive evaluation. We really appreciate that.

Best regards

Jens Fedder

Reviewer 3 Report

The current manuscript is soundly efficient to be published in its current form . The authors implemented their clinical trial perfectly and the results are presented and discussed in a clear and simple manner.

Author Response

(The authors gave the same response as above.)
